# Metabolic Profiling and Metabolites Fingerprints in Human Hypertension: Discovery and Potential

**DOI:** 10.3390/metabo11100687

**Published:** 2021-10-07

**Authors:** John Oloche Onuh, Hongyu Qiu

**Affiliations:** Center for Molecular and Translational Medicine, Institute of Biomedical Sciences, Georgia State University, Atlanta, GA 30303, USA; jonuh@tuskegee.edu

**Keywords:** metabolomics, hypertension, metabolites fingerprints, biomarkers, aging

## Abstract

Early detection of pathogenesis through biomarkers holds the key to controlling hypertension and preventing cardiovascular complications. Metabolomics profiling acts as a potent and high throughput tool offering new insights on disease pathogenesis and potential in the early diagnosis of clinical hypertension with a tremendous translational promise. This review summarizes the latest progress of metabolomics and metabolites fingerprints and mainly discusses the current trends in the application in clinical hypertension. We also discussed the associated mechanisms and pathways involved in hypertension’s pathogenesis and explored related research challenges and future perspectives. The information will improve our understanding of the development of hypertension and inspire the clinical application of metabolomics in hypertension and its associated cardiovascular complications.

## 1. Introduction

Metabolomics is the systematic study of the unique chemical fingerprints of small molecules or metabolites that are related to various cellular metabolic processes in the cell, organ, tissue, biofluids, or organism [1,2,3]. It is a relatively new field that is finding applications in all areas of science with a promising approach to discovering new biomarkers of diseases [4,5]. The metabolites, including sugars, peptides, lipids, steroids, vitamins, fatty acids, amino acids, organic acids, and numerous other small molecules, can provide comprehensive information on the physiological state and metabolic pathways; thus, it is very versatile in systems biology [1]. The metabolic profile perturbations under various environmental or physicochemical conditions also offer new insights on disease pathogenesis and treatment approaches [6]. Therefore, metabolites’ fingerprints are becoming a potent and high-throughput tool to discover biomarkers of diseases, drug toxicity, food intake, and many other processes [4,7].

Hypertension is a leading risk factor for cardiovascular disease (CVD) complications, cerebrovascular diseases, kidney failure, and death in many patients [6,8]. In 2015, the global prevalence of hypertension was reported to be 1.13 billion, suggesting a 15.4% global prevalence [9], which is projected to increase to 29.2% by 2025 [10,11]. It was once considered a problem of affluent societies, but it is increasingly posing challenges in low and middle-income countries globally due to population increases [9,12,13]. Essential hypertension (EH) is considered a multifactorial disorder resulting from a complex interplay of genetic, environmental, and lifestyle factors [10], which involves the comprehensive mechanisms associated with the perturbations in the renin-angiotensin-aldosterone systems (RAAS), oxidative stress, endothelial dysfunction, vascular tone, salt sensitivity, and sympathetic nervous system (SNS) [10,14]. Serum metabolic profiles assessed in hypertensive patients indicated that metabolic abnormalities or dysfunction, such as metabolic syndrome disorder, play a critical role in the pathogenesis of essential hypertension [6,15,16,17]. The application of metabolomics profiling or fingerprints has increased in the study of hypertension and blood pressure (BP) regulation since the first study in this field was published [6,15].

Although several treatment options are currently available for hypertension and BP control, only 53.7% of hypertensive adults receive treatments, and the success rate is put at a little over 8% [18,19]. There is, therefore, an urgent need for more proactive and effective ways to personalize drugs and other treatments in a patient-specific manner [20]. Consequently, the early detection of hypertension through biomarker discovery and related pathophysiological mechanisms holds the key to controlling the condition and preventing cardiovascular and other associated complications. Metabolomics profiles and fingerprints would help to find practical applications as a next-generation tool for proffering solutions to problems in clinical hypertension. This review summarizes the latest progress of metabolomics and metabolites fingerprints, mainly the current trends in applying metabolomics fingerprints in human hypertension. We also discuss the associated underlying mechanisms and related pathways, the potential clinical challenges, and future perspectives.

With a comprehensive search of the PubMed database, we collected the published articles on metabolomics and hypertension. Specifically, we mainly included recent studies and highlighted new information on the following aspects; first, we give an overview of metabolomics and metabolite profiling/fingerprinting. Secondly, we discuss metabolomics fingerprinting in hypertension and the associated mechanisms, focusing on the key metabolic mechanisms; glucose metabolism, amino acids metabolism, fatty acids metabolism, oxidative stress, inflammation, and steroid hormones biosynthesis. Thirdly, we provide an overview of the limitations, challenges, and future perspectives of metabolomics fingerprinting in hypertension.

## 2. Metabolomics and Metabolite Profiling/Fingerprinting

Metabolomics and metabolite profiling/fingerprinting offer a comprehensive measurement of metabolic changes resulting from responses to intrinsic or extrinsic agents, including diet, drugs, gut microbial, lifestyle, and other factors [4,21]. They are extensively used to assess metabolic changes in human health and disease and for biomarker discovery, especially in CVD, including hypertension [20,22,23]. Metabolomics has made possible the unbiased detection, identification, and quantification of several low molecular weight metabolites that are present within different matrices (cells, tissues, and body fluids) [23,24]. Since the low molecular weight metabolites are intermediates or end products of cellular metabolism that determine an organism’s functionality [4], these measurements may improve our understanding of changes in biochemical pathways that may predispose individuals to clinical manifestations of various pathologies, especially hypertension [10].

The human metabolome is made up of several thousands of small molecules (>8500), with the information contained within the metabolome being specific to each metabolite due to differences in polarity, size, and concentration [1,23,25]. This creates a considerable analysis challenge in terms of the detection, quantification, and identification of metabolites. Since it is impossible to use any single analytical method to capture all the metabolites in human samples [23], a combination of multiple techniques is usually employed to improve the understanding of the metabolite profiles in a given sample [23,26]. Several methods are currently being used, including capillary electrophoresis coupled to mass spectrometry (CE-MS), nuclear magnetic resonance coupled mass spectrometry (NMR), liquid chromatography coupled mass spectrometry (LC-MS), gas chromatography coupled to mass spectrometry (GC-MS), and Fourier transform infrared spectrometry (FTIR) [1,27]. NMR coupled to MS has been reported to be more effective and is becoming a more popular and widely applicable technique of choice [1,23]. Multivariate statistical analysis and pattern recognition tools (principal component analysis, PCA and partial least square discriminant analysis, PLSDA) are then used to compare and identify features that distinguish these complex metabolic profiles [28]. Computer-aided statistical software with unique algorithms is then used for metabolite identification by filtering, binning, aligning, and normalizing all the detected features [29]. Following this, we expect data between studies can be compared if the methods are adhered to. There is also the need for carefulness and expertise of the researcher and the sophistication of the instrumentation that goes with it and for harmonization of the different techniques for uniformity across the global space. In most instances, to better streamline the tons of information generated, the researcher sets the minimum and maximum cut-off or threshold for which data can be accepted, for instance, using ≥ 2 fold changes. Additionally, various databases and spectral libraries such as METLIN (https://metlin.scripps.edu/, accessed on 4 October 2021), Human Metabolome Database (https://hmdb.ca/, accessed on 4 October 2021), NIST (https://www.nist.gov/, accessed on 4 October 2021), and KEGG (https://www.genome.jp/kegg/, accessed on 4 October 2021) are subsequently used for biomarker identification and pathway analysis [1,29]. The various techniques, methodologies, and instrumentations used in metabolomics have been previously reviewed extensively elsewhere [4,23,29,30]. An overview of a typical metabolomics workflow from sample preparation to metabolites identification and pathways analysis is shown in Figure 1.

## 3. Metabolomic Fingerprinting in Hypertension and the Associated Mechanisms

Although the traditional and conventional methods for screening essential hypertension using a sphygmomanometer are easy and reliable, the late onset of symptoms creates the potential problem of the screening being ignored [10]. Additionally, the BP measurement cannot reveal the underlying molecular mechanisms [15], leading to the necessity of using metabolomics profiling to improve our understanding of these biochemical pathways in hypertension [10,23,31].

Metabolomics, even though relatively new, is finding applications in many science disciplines because of its high throughput and the relatively short time required for the experimental analysis [1]. Numerous metabolomics studies have been conducted to provide helpful information on the pathogenesis of several CVDs, such as cardiac hypertrophy, heart failure, coronary heart disease, and hypertension, and identify metabolites that could be considered potential biomarkers of diseases [4,23,32]. It is known that hypertension is a multifactorial disease, and as such, the application of metabolite fingerprinting to hypertension research is also multifaceted. Current research into hypertension includes investigating the effects of lifestyles and environmental factors, disease processes, and drug response [4]. Previous metabolomics studies have suggested several potential metabolic pathways responsible for hypertension, such as inflammation, oxidative stress, lipid profile, and gut microflora [4,23,33,34]. Interestingly, these pathways are also related to creating a cascade of events that may complicate the condition’s severity. Therefore, arresting any of these pathways becomes a step to reducing the onset and complications of hypertension and associated CVDs and organ damage.

It is known that circulating metabolites represent both endogenous and exogenous metabolic pathways that could provide a better understanding of diseases, especially hypertension [33]. Metabolomics effectively stratified the early molecular determinants that predispose certain individuals to early cardiovascular risk factors, such as early vascular aging (EVA), which is more prevalent among the Black population than the White population [9]. A similar difference was reported between men and women when the metabolomics profiles of 28 individuals with arterial hypertension were compared with their healthy control counterparts [35]. More interest is gradually building on applying metabolomics in clinical hypertension, with equally interesting outcomes [6,22,33,36]. The metabolic pathways associated with hypertension are discussed in the following section and illustrated in Figure 2.

### 3.1. Glucose Metabolism

The glucose metabolism has been implicated in insulin release and insulin sensitivity, which consequently influences the progression of hypertension by an increase in sodium reabsorption, SNS, and calcium concentration in vascular smooth muscle cells (VSMC) [37]. These processes increase vascular resistance and eventual increases in BP [37,38]. A cross-sectional study, including 420 consecutively referred essential hypertensive patients who were studied at 16 hypertension clinics, assessed the prevalence of glucose abnormalities in a population of patients with essential hypertension and showed that two-thirds of the patients with essential hypertension show an abnormal glucose metabolism [16]. A metabolomics study has shown that increases in glucose are associated with higher BP and EH [39]. In a metabolomics study of human serum samples from 34 hypertensive patients and age- and gender-matched healthy subjects, using GC/TOFMS-based metabolite profiling, several metabolites, including glucosamine and D-sorbitol that are associated with glucose metabolism, were identified [39]. The hypertensive patients were 14 males and 20 females aged 61 ± 13 years, while the healthy controls were 29, including 12 males and 17 females aged 59 ± 11 years. The total ion chromatogram (TIC) was also found to be different between hypertensive patients and healthy controls, with a clear separation in unsupervised principal component analysis (PCA) and orthogonal partial least square (OPLS). The metabolic profiles of the two groups also showed clear separation. In all, 38 metabolites were selected as biomarkers, including carbohydrate, lipids, amino acids, and ketones, suggesting deregulation of purine metabolism, glucose metabolism, fatty acid metabolism, amino acid metabolism, and urea cycle to be associated with hypertension. Besides glucose, other monosaccharides and disaccharides (galactose, glucosamine, sorbose, sucrose, sorbitol, inosose, and myoinositol) were all elevated in the hypertensive group. On the other hand, fructose, cellobiose, and lactobionic acid levels were decreased, suggesting that hypertension may be associated with a dysregulated carbohydrate metabolism and is accompanied by impaired glucose tolerance [39]. This was also attributed to a decline in insulin response leading to impaired insulin regulation in the catabolism of triglycerides occasioned by the increased levels of serum-free fatty acids. A recent cohort study included 4747 women whose maternal glucose values at the pregnancy OGTT were not independently associated with maternal blood pressure outcomes 10–14 years postpartum; however, insulin sensitivity during pregnancy was associated independently of blood pressure, BMI, and other covariates measured during pregnancy [40].

### 3.2. Amino Acids Metabolism

The role of amino acid metabolism in EH has been researched due to the associated biomarkers discovered in this disease. [9,34,41,42]. It was previously thought that EH might be a genetic abnormality of inherited amino acid metabolism [34,43]. In a UPLC-Q-TOF/MS-based urine metabolomics study of 75 EH and healthy control (HC) groups, ten potential biomarkers, including L-methionine, were screened as differentiating between the groups, suggesting amino acid metabolism as one of the critical pathways for the regulation of hypertension [34]. The case-control study used the following classification of BP (normal<120/80 mmHg, elevated 120–129/<80 mmHg, Hypertension Stage 1, 130–139/80–89 mmHg, and Stage 2, ≥140/≥90 mmHg). Methionine and its intermediate product, homocysteine, have been associated with CVD [34] by impairing endothelial cells’ structural and functional integrity, oxidative stress response, and vascular elasticity, aggravating the formation of atherosclerosis and numerous other conditions [34,44]. Activation of SNS also increased the secretion of catecholamine (epinephrine and norepinephrine), which induces vasoconstriction and activation of RAAS, leading to an increase in BP. Tyrosine may, therefore, be considered a biomarker of hypertension as 3,4-Dihydroxyphenylglycol (DOPEG), a precursor of catecholamine (a major neurometabolite of norepinephrine) is involved in tyrosine metabolism [34].

Metabolic profiles were also recently used to distinguish metabolic features between young Black and White adults to determine ethnic associations with BP and arterial stiffness [9]. Arterial stiffness is an independent predictor of cardiovascular disease. Increased arterial stiffness is highly associated with hypertension, especially isolated systolic hypertension [45], CVD outcomes, and mortality [9,46,47]. Arterial stiffness is found to be greater among Blacks than Whites. However, whether this difference is due to genetic dispositions or early life exposures remains uncertain [9,48,49,50]. The metabolomics study of Black and White adults (80 each) aged 20–30 years, with clinical BP <140/90 mmHg, was conducted using NMR, LC-MS, and GC-MS platforms to identify metabolites that are associated with arterial stiffness and central BP. Thirty-four metabolites were found to be significantly changing between Black and White adults; however, higher urinary non-essential amino acids were found to inversely correlate with central systolic BP (cSBP) and central pulse pressure (cPP) only in Black adults. The increased levels of non-essential amino acids were reported to be likely due to either increased protein catabolism or biosynthesis of endogenous amino acids. These non-essential amino acids were reported to be hydroxyproline, alanine, glutamine, glycine, histidine, and serine, indicating the link with collagen metabolism, glucose metabolism, and oxidative stress. This outcome also implies that upregulation of the biosynthesis of non-essential amino acids may offer protection against the onset of early vascular diseases [9]. The strength of the study lies in the fact that it consisted of a modest bi-ethnic group with participants across the arterial stiffness spectrum. However, it also included participants from the understudied Black population prone to developing EVA. It was also difficult to exclude residual confounders even though the results showed consistency after several reported adjustments.

In addition, filtered serum (lacking lipoproteins and proteins) metabolomics of 64 subjects with essential hypertension and 59 healthy controls using NMR spectroscopy identified several amino acids (alanine, arginine, and methionine), pyruvate, adenine, and uracil to be associated with EH [10]. The approach was reported to reveal subtle metabolic differences with EH. Changes in L-arginine/nitric oxide (NO), NO pathways have long been recognized to be associated with the pathogenesis of hypertension [10,51]. Although many explanations were made, the most plausible seems to be the involvement of asymmetric dimethylarginine (ADMA), which competitively inhibits NO synthase (NOS) and decreases NO bioavailability even when the concentration of L-arginine is high [10,52]. This process triggers a whole set of other physiological processes, leading to ADMA accumulation, reduced NO synthesis, and consequent increase in BP [10]. The results suggested that these metabolites could serve as biomarkers of EH and that NMR could serve as an alternative diagnostic tool in clinical settings in addition to BP monitoring. Similar studies using larger sample sizes were proposed to validate the present study’s outcome [10].

### 3.3. Fatty Acids Metabolism

There is a strong association between hypertension and abnormalities in lipid metabolism and dyslipidemia [34]. For instance, butyric acid and 5-hydroxyhexanoic acid were found to be significantly higher in the urine samples of hypertensive patients than their HC counterparts, suggesting that fatty acid metabolism may likely be associated with the pathophysiology of hypertension [34]. Saturated fatty acids are known to promote platelet aggregation, inhibit fibrous proteases, promote thrombosis, and damage vascular endothelial cells [34,53]. Interestingly, these short-chain fatty acids have become a significant focus of research in recent years because of their essential roles as primary metabolites of the intestinal flora [54,55,56].

The association of circulating metabolites with longitudinal BP changes was recently investigated and validated in two different cohorts (Prospective Investigation of the Vasculature in Uppsala Seniors, PIVUS, and Uppsala Longitudinal Study of Adult Men, ULSAM, respectively) using LC and GC coupled with MS [33]. Circulating metabolites levels with BP changes at baseline, clinical BP stage, and a follow-up period of 5 years later they were assessed to determine their association. Of the five metabolites associated with BP change in the discovery cohort, two, diacylglycerol and monoacylglycerol, were validated to be associated with diastolic BP change, suggesting a pathophysiological pathway for these metabolites in hypertension [33]. The association of ceramide, triacylglycerol, total glycerolipids, and oleic acid) with longitudinal change in DBP was reported to confirm their potential significance in the development of hypertension. Ceramide was considered to be the central intermediate in the sphingolipid biosynthetic pathway and mediates vascular dysfunction via inhibition of endothelial NOS-serine/threonine protein kinases heat shock protein 90 signaling complex [33]. It also causes endothelium-dependent contraction by induction of the release of the pro-inflammatory compound, thromboxane A2, via the calcium-independent phospholipase A2 (iPLA2), cyclooxygenase-1 (COX-1), and thromboxane synthase (TXAS). The triacylglycerols were reported to associate with longitudinal change in DBP through insulin resistance. The association of fatty acids, such as oleic acid, with longitudinal change in DBP was attributed to increases in mitochondrial ROS generation. However, this outcome is contrary to observations in previous studies in which the oleic acids were suggested to play a vasodilating effect, thereby causing a reduction in blood pressure [57,58]. A recent study also showed the protective action of oleic acid against cellular lipotoxicity, attenuating oxidative stress, endoplasmic reticulum stress, inflammation, and apoptosis [59]. Understanding these pathways could lead to a better knowledge of the treatment of the causes of hypertension and improved health outcomes. The strength of the study lies in it having a well-defined population and correction for multiple testing, and potential confounders were adjusted appropriately. However, the study was limited by the use of only elderly Swedish adults and so cannot be generalized to other populations and age groups; The generalizability was also determined by excluding the persons with comorbidities and the use of antihypertensive medications [33].

Although metabolic abnormalities have long been indisputably known to be one of the significant causes of hypertension, not much is known about the associated biomarkers and the pathways through which they are mediated [6,33]. A targeted metabolomics study was conducted to detect potential biomarkers of hypertension in plasma samples of EH patients and healthy subjects and to evaluate the effects of acupuncture using multiple reaction monitoring mass spectrometry (MRM-MS) [6]. Untargeted metabolomics is the defined as the “comprehensive analysis of all the measurable analytes in a sample, including chemical unknowns,” while targeted metabolomics is defined as the “measurement of defined groups of chemically characterized and biochemically annotated metabolites” [60]. Of the 47 compounds screened as target metabolites, 2 of the compounds, oleic acid, and myoinositol, were considered the most distinguishing metabolites between plasma samples of hypertensive patients and healthy subjects, closely correlated with 24 h BP and nocturnal dipping. However, treatment with acupuncture was able to return the levels to normal, with a corresponding reduction in BP and improvement in circadian BP rhythm, demonstrating the potential roles of these metabolites as biomarkers of hypertension [6]. However, this relationship of oleic acid with 24 h BP changes needs further investigation, as oleic acid was previously reported to have the vasodilating effect [57,58].

In addition to EH, fatty acid metabolites with an increased fatty acid metabolism have also been implicated in pulmonary arterial hypertension (PAH). This is accompanied by the alterations in fatty acid oxidation and disruptions in both glycolytic pathways and the tricarboxylic acid (TCA) cycle [61]. These results suggested a specific metabolic pathway for vascular remodeling, which is modulated through transcriptional control of the enzymes. However, fatty acid oxidation was reported to be a more efficient process compared to glycolysis. It was reported that the identified metabolites could act as possible biomarkers for the diagnosis of the condition and also open opportunities for clinical treatments targeting the metabolic pathways leading to PAH [61]. Additionally, inhibition of glycolysis at the early onset of PAH and the inhibition of fatty acid oxidation at the advanced stage of the disease may help treat the condition [61].

An untargeted plasma metabolomics approach was recently used in a case-control study to understand the metabolomics fingerprints of EVA syndrome in hypertension and identify metabolites that could be used for prognosis and/or early diagnosis of the disease [32]. Since arterial stiffness, a predictor of vascular aging, is associated with hypertension, decreasing arterial stiffness would reduce BP and improve patients’ survival from other CVD risk factors [32,62,63,64]. The results obtained from age-, BMI- and sex-matched groups of EVA and non-EVA individuals with hypertension from the CARE NORTH prospective cohort characterized 497 metabolites, out of which four identified as lysophosphatidylcholines (LPCs) (LPC 18:2, LPC 16:0, LPC 18:0 and LPC 18:1) were found to be independently associated with EVA. However, it is unknown whether these LPCs are playing a causal role or are a consequence of other compensatory mechanisms. Since LPCs are known to act as regulators of oxidative stress in the aging aorta [32,65,66], these findings provide a new understanding of the metabolomics manifestation of vascular aging and can serve as a potential biomarker and predictor of EVA in hypertensive patients that could be useful in the treatment and personalized care for CVD [32].

### 3.4. Oxidative Stress

Several studies have established a strong association between oxidative stress and EH [34,67,68,69]. Oxidative stress has multiple effects on hypertension, such as causing vascular endothelial dysfunction due to NO depletion, stimulation of vascular renin-angiotensin system (RAS), angiotensin II production, the proliferation of vascular smooth muscle cells, and the generation of reactive oxygen species (ROS) [34,70]. These processes subsequently cause vascular constriction, increased BP, acceleration of arteriosclerosis, and hypertension development [34,69,70].

The association of amino acid metabolism with oxidative stress in EH has been reported. It was previously reported that metalotin acts as a powerful antioxidant with the ability to inhibit oxidative stress, especially in the endothelium, increasing vasodilation and reducing BP, improving cardiovascular function [71,72]. Metalotin was found to be significantly higher in the urine samples of hypertensive patients than their HCs, suggesting that it may be acting as an antioxidant [34]. Besides, a dried blood spot metabolomics analysis was adopted to detect metabolites from 87 essential hypertension and 91 healthy controls [31], and found that glycine, ornithine, C10 (decanoylcarnitine), Orn/Cit, Phe/Tyr, and C5-OH/C8 (3-hydroxyisovalerylcarnitine/octanoylcarnitine) were different between groups, suggesting their potential to be used as biomarkers for diagnosing essential hypertension [31]. It was reported that amino acid glycine reduces oxidative stress by regulating the biosynthesis of glutathione, improves the activities of antioxidant enzymes, inhibits inducible nitric oxide synthase (iNOS) expression, and alters endothelium-dependent relaxation [31].

The role of LPCs in the activation of oxidative stress is well known [32,65,66]. It has been shown to enhance oxidative stress in the aging rat aorta by producing reactive species and activation of the 5-lipoxygenase pathway [32,66]. However, the results appear contradictory. For example, some studies showed a positive association in the aortas of apolipoprotein E knockout mice during early stages of atherosclerosis [64]; in contrast, others showed inverse associations with carotid-femoral pulse wave velocity (cfPWV), heart rate, ADMA, and ADMA/arginine ratio in atherosclerotic patients compared to controls [73]. Lower LPC levels were found to be associated with EVA and hypertension. However, this area may warrant further investigation and clinical studies to resolve these contradictions in research outcomes [32]. It will help to establish the cause-effect relationship between LPCs and EVA and hypertension [32].

### 3.5. Inflammation

It has been established that low-grade inflammation could be associated with the initiation and maintenance of hypertension due to the modulation of both innate and adaptive responses [36,74]. Several factors may contribute to the inflammation, particularly higher levels of phosphatidylcholines (PCs) and LPCs. PCs are converted into LPCs by a pro-inflammatory mediator, phospholipase A2 (PLA2) [36,75]. LPCs have also been implicated in atherosclerosis and other inflammatory diseases by altering various physiological functions in different cell types [76]. Additionally, increased fatty acids, acylcarnitines, and insulin resistance play major roles in activating pro-inflammatory signaling pathways, contributing to the rapid onset of hypertension [77,78,79]. Endothelial AMP-activated protein kinase (AMPK) has also been known to be associated with vascular homeostasis and inflammation, making it a novel target for the treatment of pulmonary arterial hypertension (PAH) [80].

A serum metabolomics study on a healthy UK population was recently conducted to identify biomarkers and pathways associated with hypertension and dyslipidemia by comparing their respective metabolic patterns using GC-MS and ultra-performance LC-MS technologies [36]. Several metabolites (26 and 46) were found to be both common and distinct to hypertension and dyslipidemia, respectively, suggesting that hypertension and dyslipidemia have independent and synergetic biological significances [36]. The metabolites associated with fatty acids metabolism, glycerophospholipid metabolism, and amino acid metabolism were found to be involved in major proinflammatory pathways that are known to cause insulin resistance, vascular remodeling, macrophage activation, and oxidized low-density lipoprotein (LDL) formation [32,36].

A disruption of LPCs is reported to highlight their potential role in inflammation [35]. This is because the concentration of blood PCs may also be associated with several other comorbidities of hypertension, especially BMI, dyslipidemia, and insulin resistance [35,81]. While it was observed that this metabolite is common to both men and women, it was, however, reported that the overrepresentation of the acyl-alkyl forms of the PCs was characteristic to only men but not women [35]. The increased concentration of these metabolites, previously called plasmalogens, could suggest an alternative antioxidant defense system for men [35].

Chronic inflammation is known to be one of the major manifestations of EVA, along with increased arterial stiffness, endothelial dysfunction, impaired vasodilatation, and dyslipidemia [32,82]. An untargeted metabolomics profile of plasma metabolites in a case-control study of EVA and non-EVA individuals with hypertension revealed LPCs associated with EVA, hypertension, and inflammation [32]. The association of LPCs with chronic inflammation, EVA, and hypertension is not surprising, considering the role it plays in oxidative stress, atherosclerosis, and coronary disease. It is a significant part of the phospholipids of oxidized LDL and has been implicated in cell proliferation and migration, inflammation, and oxidative stress [83,84].

### 3.6. Steroid Hormones Biosynthesis

Steroid hormone biosynthesis has also been reported to be one of the main pathways responsible for the pathogenesis of hypertension [34]. Increased levels of these hormones have been associated with a higher risk of hypertension due to increased sodium absorption by regulating renal mineralocorticoid and glucocorticoid receptors [34,85,86]. The levels of hydroxyandrosterone (a product of androsterone) and cortolone were significantly elevated in hypertensive patients compared to HCs, suggesting that steroid hormone biosynthesis was associated with hypertension [34]. Androsterone has previously been reported to have the ability to reduce the micro-viscosity of the cell membrane and, consequently, red blood plasticity, causing the red blood cells to form large clots leading to the development of proliferation and hypoxia with a significant effect on the cardiomyocytes and BP [34,87]. Other hormones, including epiandrosterone sulfate, 5 α-androstan-3β-diol disulfate, androsterone sulfate, melatonin cortolone, dihydroxyphenylglycol, and hydroxyandrosterone, were also reported to play a notable role in arterial hypertension [35]. Identifying these biological signatures in clinical hypertension patients using metabolomics platforms very early in the process will significantly reduce the burden of hypertension and its associated cardiovascular complications.

Although the individual metabolic aspect has been discussed above separately, it is notable that these diverse metabolic profiling systems are likely modulated simultaneously by some common mechanisms, and multiple cross-talks exist among these pathways, playing a synergic effect on the development of hypertension. For example, many studies have shown that AMP-activated protein kinase (AMPK) participated in cellular function by regulating the multiple branches of metabolism, including glucose, fatty acid and amino acids’ metabolisms, oxidative stress, and inflammatory response, providing a comprehensive regulatory mechanism in hypertension. Accordingly, a number of pharmacological compounds that increase AMPK activity directly or indirectly have been identified and have shown positive effects on many aspects of cardiometabolic disease, including hypertension. A small number of these compounds have also been approved for use or are in clinical trials, such as resveratrol, berberine, metformin, salicylate, and their antihypertensive effects may be partially mediated in part by AMPK [88]. In addition, the sodium/glucose cotransporter 2 (SGLT2) inhibitor canagliflozin, which has recently been approved for treating type 2 diabetes, also showed effects of indirectly activating AMPK [88] and improving the metabolic profile and cytokines [89]. These signs of progress bring new insights into the therapeutic potential in hypertension.

## 4. Limitations, Challenges, and Future Perspective of Metabolomic Fingerprinting in Hypertension

Although some advances have been made in recent years in applying metabolomics techniques, this cannot be said to be true regarding clinical hypertension [34]. One of the challenges is the complexity of the organism and individual variations in human hypertension that makes it extremely difficult to go beyond basic research in animal models and some few human studies utilizing blood for biomarker discovery [10,34]. Secondly, species differences between humans and rodents lead to inconsistent results, which also poses a significant challenge to the successful application of metabolomics profiling in clinical hypertension [33,90]. A combination of basic laboratory measurements with comprehensive clinical methods of assessment, such as in metabolomics, will reveal many metabolic perturbations closely associated with BP increases and hypertension [6]. Metabolomics approaches, if used properly, could provide better insights into metabolic changes occurring in hypertension which could be beneficial to our understanding of associated metabolic pathways and assist in the development of biomarkers for early diagnosis and treatment of hypertension [31]. In addition, most of the reported clinical studies were small, cross-sectional studies and used a targeted metabolomics approach that may have been biased and not captured the entire metabolite profiles that can predict metabolic perturbations and discriminatory metabolites at any given time [4,23]. Moreover, even among humans, differences in ethnicity, diet, and gender can affect metabolite profiles [9]. More extensive clinical studies with large sample sizes are therefore needed for the practical application of metabolomics in clinical settings [36]. Further development of this field, especially in terms of instrumentations, data analysis, and rapid detection, could lead to the detection and identification of more endogenous metabolites that can be used for different physiological processes involved in clinical hypertension [34]. This will significantly reduce and cut down the time, cost, and other associated constraints that may militate against the adoption of these novel techniques and approaches.

While metabolomics profiling has tremendously improved our knowledge of disease conditions in the last ten years, its importance in science and health as a valuable tool for diagnosing health complications is becoming more widely accepted; a lot remains to be achieved to give it the necessary attention that it deserves. Metabolomics relies heavily on the development of appropriate technology for biological research, which can also be costly. Any rapid drop in technology cost will go a long way in making the metabolomics approach readily available for broader applicability [91]. Another critical aspect that should not be ignored is metabolomics data storage and analysis. Therefore, there is the need for all specialists in the field and other professionals to work collectively to ensure continuous development and improvement of this area [91]. It is also vital to consider the economics and cost–benefits of metabolomics profiling associated with personalized medicine, which should be based on a comprehensive health economic analysis to improve the cost of health and better delivery systems. The integration of metabolomics data with genomics and other “omics” technologies, such as proteomics, peptidomics, and transcriptomics, could provide opportunities for clinicians and researchers to better understand the pathophysiology of diseases, especially in CVDs and hypertension associated with inter-individual variations in response to given exposures to predisposing conditions [92]. This could be in the form of risk prediction by developing real-time monitoring of patients for easy detection of biomarkers that can predict either improvement or worsening of the condition and early intervention targeting.

## 5. Conclusions

The application of metabolomics in clinical hypertension, though relatively new, is rapidly growing, with great promise to transform patient treatment and care. It is a potent and high throughput tool that offers new insights on disease pathogenesis and proffers more informed and targeted treatment approaches. Metabolomics profiling makes possible the unbiased detection, identification, and quantification of several low molecular weight metabolites present within different matrices that could be considered biomarkers of hypertension. This also helps our understanding of the associated pathways through which hypertension could be regulated. Some of these potential pathways identified through metabolomics profiling are inflammation, amino acid metabolism, oxidative stress, lipid profile, steroid hormones, and glucose metabolism. As illustrated in Figure 3, these pathways are interwoven, creating a cascade of events that complicates the pathogenesis of hypertension. Impairment of metabolism or hormone disorder activates pro-inflammatory signaling pathways and increases oxidative stress, causing insulin resistance, macrophage activation, oxidized LDL formation, and impaired vasodilatation, consequently resulting in vascular cell proliferation and migration, endothelial dysfunction, and vascular remodeling, leading to EVA and hypertension. Based on the data presented here, it is expected that metabolites of amino acid metabolism (arginine, glycine and NO), fatty acid metabolism (LPCs) and associated oxidative stress metabolites might be developed as promising biomarkers. More interestingly, it helps in the stratification and differentiation of these metabolites and pathways by ethnicity, sex, and age, thereby better improving our knowledge of some underlying causes or predisposing factors of hypertension. This will significantly assist in reducing the burden of hypertension and its associated cardiovascular complications. All the relative studies human hypertension that have been discussed in this article are summarized in Table 1.

## Figures and Tables

**Figure 1 metabolites-11-00687-f001:**
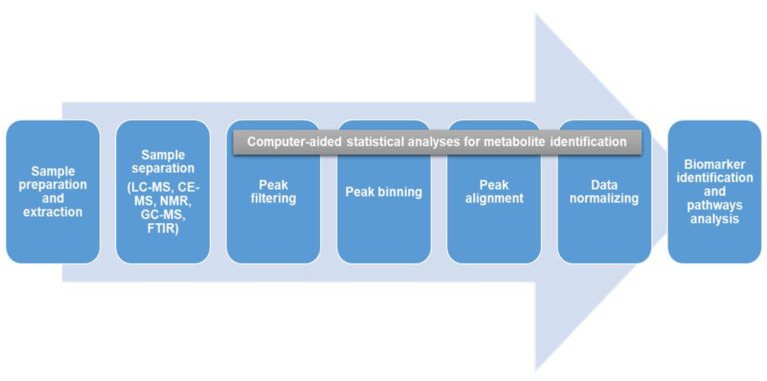
An overview of a typical metabolomics workflow from sample preparation to biomarker identification and pathways analysis. A combination of multiple techniques is often employed to detect, quantify and identify metabolites. Computer-aided statistical software is used to identify these metabolites by filtering, binning, aligning and normalizing detected features, and the pathways are identified using various databases and spectral libraries.

**Figure 2 metabolites-11-00687-f002:**
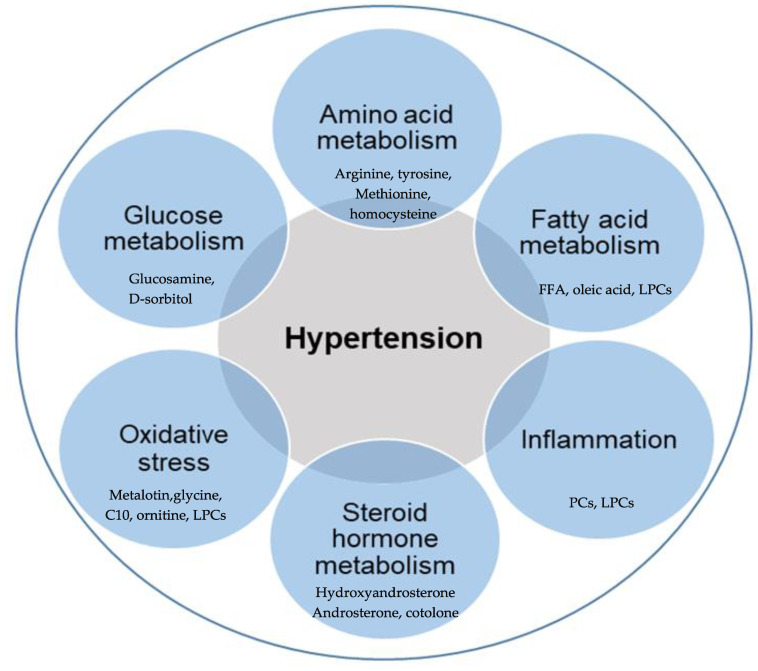
Overview of metabolic processes associated with hypertension. Metabolites are associated with hypertension, mainly resulting from the dysregulation of various metabolisms of amino acid, fatty acid, glucose, and steroid hormone, increased oxidative stress, and chronic inflammation. FFA (free fatty acids), PCs (phosphatidylcholines), LPCs (lysophosphatidylcholines), C10 (decanoic acid).

**Figure 3 metabolites-11-00687-f003:**
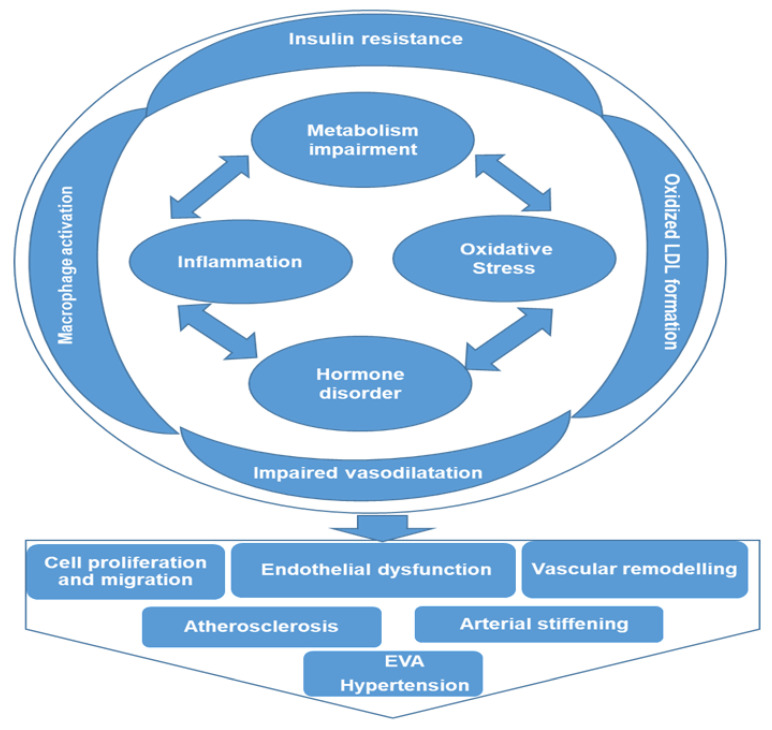
Potential comprehensive metabolic mechanisms associated with the initiation and maintenance of hypertension. Multiple signaling pathways are involved in metabolic mechanisms associated with hypertension, and these pathways are interwoven, creating a cascade of events that complicates the pathogenesis of hypertension. Impairment of metabolism or hormone disorder activates pro-inflammatory signaling pathways and increases oxidative stress, causing insulin resistance, macrophage activation, oxidized LDL formation, and impaired vasodilatation, consequently resulting in vascular cell proliferation and migration, endothelial dysfunction and vascular remodeling, leading to early vascular aging (EVA) and hypertension.

**Table 1 metabolites-11-00687-t001:** Summary of human studies on metabolomics and hypertension.

S/N	Study	Methods	Results	References
1	A targeted metabolomics MRM-MS study on identifying potential hypertension biomarkers in human plasma and evaluating acupuncture effects	Multiple reaction monitoring—mass spectrometry (MRM-MS) of plasma samples from healthy and hypertensive patients	Forty-seven (47) chemical entities were detected in plasma samples of patients. Oleic acid and myoinositol were strongly correlated	[6] Yang et al., 2016
2	Central systolic pressure and a nonessential amino acid metabolomics profile: the African Prospective study on the Early Detection and Identification of Cardiovascular disease and Hypertension	NMR spectroscopy, liquid chromatography-tandem mass spectrometry and gas chromatography–time of flight-mass spectrometry of plasma and urine samples	Thirty four metabolites were to be differentiated between Black and White groups. cSBP and cPP inversely correlated with various non-essential amino acids only in Blacks	[9] Mels et al., 2019
3	Essential hypertension: A filtered serum-based metabolomics study	NMR metabolomics of filtered serum from 64 essential hypertension (EH) patients and 59 healthy controls (HC)	Alanine, arginine, methionine, pyruvate, adenine, and uracil were found to correctly classify 99% of cases from HC, which also correlated in both isolated elevated DBP as well as combined elevated systolic-diastolic blood pressure	[10] Ameta et al., 2017
4	Application of chemometrics to ^1^H NMR spectroscopic data to investigate a relationship between human serum metabolic profiles and hypertension	^1^H NMR spectroscopy of serum profiles of patients with low/normal, borderline and high SBP	The study distinguished low/normal SBP serum samples from borderline and high SBP samples; however, borderline and high SBP samples were not distinguishable from each other. Serum metabolic profiles correlated with SBP, which was attributed to lipoproteins	[15] Brindle et al., 2003
5	Effects of four different antihypertensive drugs on plasma metabolomic profiles in patients with essential hypertension	Ultrahigh performance liquid chromatography-mass spectrometry of plasma samples from 313 hypertensive Finnish men	BP decreases correlated with decreases in long-chain acylcanitines (amlodipine and losartan), medium and long-chain FAs (bisoprolol), and an increase in plasma uric acid levels and urea metabolites (hydrochlorothazide)	[20] Hitunen et al., 2017
6	Metabolomic study for essential hypertension patients based on dried blood spot mass spectrometry approach	Dried blood spot method coupled with direct infusion mass spectrometry (MS) metabolomic analysis of 87 essential hypertension (EH) patients and 91 healthy controls (HC)	Gly, Orn, C10, Orn/Cit, Phe/Tyr, and C5-OH/C8 were reported to be key metabolites that differentiated EH patients from HC individuals and can be considered biomarkers for hypertension	[31] Bai et al., 2018
7	Metabolomic signature of early vascular aging (EVA) in hypertension	Untargeted metabolomic approach of plasma samples of age-, BMI-, and sex-matched groups of EVA (*n* = 79) and non-EVA (*n* = 73) individuals with hypertension	Four metabolites lysophosphatidylcholines (LPCs), LPC 18:2, LPC 16:0, LPC 18:0 and LPC 18:1 were associated with EVA. Hypertensive patients with the 4 downregulated LPCs had 3.8 higher risk of EVA compared to those with upregulated LPCs	[32] Polonis et al., 2020
8	Global plasma metabolomics to identify potential biomarkers of blood pressure progression	Liquid- and gas-chromatography coupled to mass spectrometry of individuals not on BP-lowering medication at baseline and followed up 5 years later	In the cohort group, ceramide, triacylglycerol, total glycerolipids, oleic acid, and cholesterylester were correlated with DBP change. In the validation cohort, diacylglycerol (36:2) and monoacylglycerol (18:0) were associated with DBP change.	[33] Lin et al., 2020
9	Identification of essential hypertension biomarkers in human urine by non-targeted metabolomics based on UPLC-Q-TOF/MS	Ultra performance liquid chromatography coupled with quadrupole time-of-flight mass spectrometry (UPLC-Q-TOF/MS) metabolomics of urine samples from 75 cases from each group of EH and HC	Ten potential biomarkers including L-methionine representing amino acid metabolism, fatty acid metabolism, steroid hormone biosynthesis and oxidative stress were found to differentiate between EH and HC groups.	[34] Zhao et al., 2018
10	Sexual dimorphism of metabolomic profile in arterial hypertension	Targeted plasma metabolomic profiles of 28 individuals (13 women and 15 men) with essential arterial hypertension and 36 HC (18 women and 18 men)	Twenty-nine metabolites were found to discriminate the metabolic sexual dimorphism of hypertension. These metabolites are related to phospholipidic and cardiac remodeling, arginine/nitric oxide pathway and antihypertensive and insulin resistance mechanisms	[35] Goita et al., 2020
11	Metabolomic characterization of hypertension and dyslipidemia	Serum metabolomics of healthy UK population using gas chromatography–mass spectrometry and ultraperformance liquid chromatography–mass spectrometry approach	The study identified 26 and 46 metabolites considered as potential biomarkers of hypertension and dyslipidemia, respectively, which were associated with the metabolisms of fatty acid metabolism, glycerophospholipid metabolism, alanine, aspartate and glutamate	[36] Ke et al., 2018
12	An ultrasonication-assisted extraction and derivatization protocol for GC/TOFMS-based metabolite profiling	A gas chromatography/time-of-flight mass spectrometry(GC/TOFMS) of human serum samples EH and HC individuals	Identified metabolite markers that were associated with hypertension to innclude glucosamine, D-sorbitol, 1-stearoylglycerol, and homocysteine	[39] Liu et al., 2011
13	Metabolomic heterogeneity of pulmonary arterial hypertension	A combination of high-throughput liquid-and-gas-chromatography-based mass spectrometry metabolomics of human lung tissue from 8 normal and 8 pulmonary arterial hypertension patients	Metabolites revealed disrupted glycolysis, increased TCA cycle, and fatty acid with altered oxidation pathways in the human PAH lung suggesting specific metabolic pathways contributing to increased ATP synthesis responsible for the vascular remodeling process in severe pulmonary hypertension	[61] Zhao et al., 2014
15	An untargeted metabolomics study of blood pressure: findings from the Bogalusa Heart Study	Untargeted, ultrahigh performance liquid chromatography-tandem mass spectroscopy metabolomics profiling among 1249 BHS participants	A total of 24 novel metabolites robustly associated with BP including 3 amino acid and nucleotide metabolites, 7 cofactor and vitamin or xenobiotic metabolites and 10 lipid metabolites and their various metabolic pathways	[79] He et al., 2020
14	Top-down lipidomics reveals ether lipid deficiency in blood plasma of hypertensive patients	Plasma lipidomics study of 19 hypertensive males and 51 normotensive male controls using top-down shotgun profiling on a LTQ Orbitrap hybrid mass spectrometer	Plasma of hypertensive individuals had decreased content of ether lipids. Ether phosphatidylcholines and ether phosphatidylethanolamines comprising arachidonic (20:4) and docosapentaenoic (22:5) fatty acid moieties, were more diminished as well as content of free cholesterol	[81] Graessler et al., 2009

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
