# Peer review of "Metabolic Profiling and Metabolites Fingerprints in Human Hypertension: Discovery and Potential"

_metabolites, 2021, doi:10.3390/metabo11100687_

Round 1

Reviewer 1 Report

The manuscript titled " Metabolic profiling and metabolites fingerprints in human hypertension: discovery and potential" is  a very interesting review that describes the role of some metabolic pathways in cardiovascular diseases in humans. The review is well written, the overall structure is appropriate and references are updated and of good quality. Authors should:

1) describe the role of PI3K-AMPK in hypertension and other CDV in preclinical model

2) describe in introduction or discussion the role of some drugs ( like SGLT2 inhibitors) able to improve the metabolic profile and cytokines involved in several CDVs and cardiotoxicity of anticancer treatments ( you can cite 10.1186/s12933-021-01346-y )

3) describe in a appropriate manner the role of some cytokines in pathogenesis of CDVs and cardiotoxicity of anticancer drugs ( cytokines plays a key role in the metabolic profile of CDVs; you can cite 10.3390/jpm10040179 )

After these changes, the review could be acceptable for publication in this journal

Reviewer 2 Report

In this review 'Metabolic profiling and metabolites fingerprints in human hypertension: discovery and potential', Onuh and Qiu summarized the progress of metabolomics and metabolite in hypertension's pathogenesis. The review is well organized and well written, and is an important contribution in the field of metabolomics and hypertension. Some minor revisions are recommended. 

1) The authors should mention briefly what metablomics techniques and analysis tools are used when discussing some important studies. Such as in glucose metabolism. 

2) Is ref.39 the only metabolomics study on glucose metabolism and hypertension? 

3) There is a grammar error in Line 200-202, please check. 

4) Line 162, should be 'A metabolomics study'; 

5) In the future perspective part, the authors should discuss a little more about the future development of metabolomics technologies and data analysis/validation tools.

6) The combination of different 'omics' may be critical to better understand a complex disease such as hypertension. The authors might want to discuss a little in the perspective. 

Reviewer 3 Report

1) Lines 266-269:The fatty acids such as oleic acid found to be associated with longitudinal change in DBP was attributed to increases in mitochondrial ROS generation. Understanding these pathways could lead to a better knowledge of the treatment of the causes of hypertension and improved health outcomes”.

Please note that there is also a lot of literature that says otherwise, oleic acid being a component with a vasodilator effect and reducing oxidative stress. Please, search for literature and add that information. That paragraph offers a pretty biased claim.

2) Lines 284-290:Of the 47 compounds screened as target metabolites, 2 of the compounds, oleic acid and myoinositol, were considered the most distinguishing metabolites between plasma samples of hypertensive patients and healthy subjects closely correlated with 24-hour BP and nocturnal dipping. However, the treatment effect of acupuncture was able to return the levels to normal with a corresponding reduction in BP and improvement in circadian BP rhythm, demonstrating the potential roles of these metabolites as biomarkers of hypertension [6]”.

This paragraph is related to the previous one. Oleic acid is one of the components that is always found in greater quantity in food and in body composition. A greater association in hypertension does not imply that it is one of the causes of this pathology. If there is evidence that they are part of coadaptive mechanisms to compensate for changes in blood pressure and the concomitant effects of metabolic syndrome. Please look for information and indicate it in the discussion.

3) Figure 2. Please indicate the description of each abbreviation at the bottom of the figure.

Round 2

Reviewer 3 Report

The reviewer thanks the authors for correcting the biases raised.